# Enhanced Non-Invasive Diagnosis of Female Urinary Incontinence Using Static and Functional Transperineal Ultrasonography

**DOI:** 10.3390/diagnostics14222549

**Published:** 2024-11-14

**Authors:** Milosz Pietrus, Kazimierz Pityński, Maciej W. Socha, Iwona Gawron, Robert Biskupski-Brawura-Samaha, Marcin Waligóra

**Affiliations:** 1Department of Gynecology and Oncology, Faculty of Medicine, Jagiellonian University Medical College, 31-501 Kraków, Poland; milosz.pietrus@uj.edu.pl (M.P.); kazimierz.pitynski@uj.edu.pl (K.P.); 2Department of Perinatology, Gynecology and Gynecologic Oncology, Faculty of Health Sciences, Collegium Medicum in Bydgoszcz, Nicolaus Copernicus University, 85-821 Bydgoszcz, Poland; 3Clinic of Gynecological Endocrinology, Faculty of Medicine, Jagiellonian University Medical College, 31-501 Kraków, Poland; iwona.gawron@uj.edu.pl; 4Department of Obstetrics, Perinatology and Neonatology, Center for Postgraduate Medical Education, Marymoncka St. 99/103, 01-813 Warsaw, Poland; robertsamaha@gmail.com; 5Pulmonary Circulation Centre, Department of Cardiac and Vascular Diseases, Faculty of Medicine, Jagiellonian University Medical College, John Paul II Hospital in Kraków, 31-022 Kraków, Poland; 6Center for Innovative Medical Education, Department of Medical Education, Faculty of Medicine, Jagiellonian University Medical College, 30-688 Kraków, Poland

**Keywords:** stress urinary incontinence, pelvic floor ultrasound, transperineal ultrasound

## Abstract

**Background/Objectives**: To investigate the utility of transperineal ultrasound in detecting stress urinary incontinence (SUI) and identify optimal anatomical and functional parameters. **Methods**: Thirty-four women presenting with SUI with or without pelvic organ prolapse between 2012 and 2016 were studied. The control group included patients without SUI who underwent surgery for mild gynecologic disorders or pelvic organ prolapse. The relationship between selected ultrasound parameters and SUI was determined. **Results**: Among the 20 variables measured in ultrasonography using 4 angles and the bladder–symphysis distance (BSD) values, we found that the difference in the BSD obtained at rest and during the Valsalva maneuver (odds ratio [OR]: 1.15, 95% confidence interval [CI]: 1.05–1.27, *p* = 0.004), the mean urethral diameter (UD; OR: 4.29, 95% CI: 2.07–8.83, *p* = 0.0001), and the occurrence of the funneling sign during the Valsalva maneuver (OR: 21; 95% CI: 6.1–71.9, *p* < 0.0001) were associated with urinary incontinence in the logistic regression analysis. The optimal cut-off point for BSD was >8 mm (area under the curve (AUC), 0.71; sensitivity, 91.2%; specificity, 56.8%; *p* = 0.001) and that for UD was >6 mm (AUC, 0.84; sensitivity, 82.1%; specificity, 73%; *p* < 0.001). **Conclusions**: Transperineal ultrasonography is a useful tool for detecting SUI. Our findings highlighted the utility of several sonographic parameters, mainly the urethral diameter, in the diagnosis of urinary incontinence.

## 1. Introduction

Stress urinary incontinence (SUI) is a common condition appearing in women. Pelvic organ prolapse (POP) and SUI may often coexist and may show inter-relationships among symptoms. The postpartum incidence of POP and SUI can be as high as 50%.

Although medical history assessments and physical examinations play key roles in the diagnosis of SUI, the use of imaging techniques and functional modalities (urodynamic testing) for diagnosing this condition has been growing since these techniques allow precise evaluations of the damage to the supporting structures that are responsible for SUI and the descent or prolapse of pelvic organs, thereby improving the possibility of therapeutic success.

The ultrasound units currently used in daily obstetrics and gynecology practice can provide insights regarding pelvic floor structures, supplementing the results of physical examinations. These ultrasound assessments can visualize the three pelvic compartments in female patients, and the images thus obtained can be used to assess the positions of anatomical structures in relation to the pelvic bones and pre-set planes and to measure the distances among these structures and predefined angles. These assessments can also provide additional insights into existing defects, allowing the targeted correction of damage and potentially reducing the rates of revision surgeries. Pelvic floor ultrasound assessments can be performed using transabdominal or transvaginal probes, both of which allow evaluations of urethral mobility with good repeatability and reproducibility [1].

Therefore, this study aimed to assess the usefulness of transperineal ultrasound examinations in detecting SUI, specifically by identifying the optimal anatomical and functional parameters and determining cut-off values for these ultrasonographic parameters.

## 2. Materials and Methods

### 2.1. Patient Enrollment

Patients hospitalized at the tertiary gynecology (Department of Gynecology and Oncology at Jagiellonian University Meical College) center between 2012 and 2016 who required the surgical management of SUI with or without POP were consecutively and prospectively enrolled in this study. Patients without urinary incontinence (UI) who were referred for gynecologic surgery due to mild gynecological disorders or POP surgery were included in a control group. We excluded patients who had previously undergone surgery for either UI or POP, patients with pelvic tumors that could affect pelvic organ statics, and those who did not provide informed consent to participate in the study. The presence of SUI was confirmed on the basis of a questionnaire (Appendix A. Patient-Reported Outcomes and Health History Questionnaire).

The study was registered in the Protocol Registration and Results System database (ClinicalTrials.gov; registration no. NCT05912491).

### 2.2. Ultrasound Measurements

Transperineal ultrasound evaluations were performed using Voluson E6 (GE Medical systems, Milwaukee, WI, USA). A latex barrier was used to secure the abdominal probe, which was positioned between the labia minora, the clitoris, and the anus. The probe was oriented to coaxially visualize the pubic symphysis on one side and the anus and rectum on the other side of the two-dimensional image. All measurements were performed in three states: during pelvic muscle relaxation, during the Valsalva maneuver, and during perineal squeezing.

The quantitative parameters evaluated in this study were as follows:Bladder–symphysis distance (BSD)—the distance between the neck of the urinary bladder and the symphysis;Alpha angle—the angle between the proximal urethral axis and the *x*-axis of the symphysis pubis (central line);Beta angle—the angle between the lines parallel to the proximal urethra and the distal axis of the urethra;Gamma angle—the angle between the lower margin of the symphysis pubis and the neck of the urinary bladder;Retrovesical angle (RVA)—the angle between the proximal urethral axis and the line tangential to the lowest part of the posterior wall of the urinary bladder;Mean urethral diameter—the sum of urethral diameters at points on the proximal, central, and distal parts divided by 3.The qualitative parameters were as follows:The total rotation urethral sign was defined as a significant disappearance of the characteristic hypoechogenic image of the inside of the urethra during the Valsalva maneuver;The funneling sign was defined as opening of the proximal urethra during the Valsalva maneuver or cough.

### 2.3. Statistical Analysis

Power analysis, based on an assumed effect size of 0.6 to 0.7, indicated that a sample size of 45 to 33 participants per group would be sufficient to detect moderate-to-large differences with a power of 0.8 and an alpha level of 0.05.

Mean values and standard deviations are used to present continuous variables, while numbers and percentages are used to present qualitative variables. The non-parametric Mann–Whitney U test was used to compare measurements in the study groups. For qualitative variables, the chi-square test was used to compare the frequency of the assessed features between groups. We used Student’s *t*-test for continuous variables, all of which had a close to normal distribution, and the chi-square test for frequency comparisons in UI patients versus the control group in terms of the ultrasound parameters used to assess POP and UI. A significance level of *p* < 0.05 was assumed throughout the study.

A series of multiple logistic regression models were used to assess the relationships between various functional and static parameters in ultrasonography. These were divided into three groups: anatomical and functional angles and diameters, urethral diameters, and sonographic symptoms at rest and after the Valsalva maneuver. Variables were selected for logistic regression if they showed significant differences in the groups with and without UI in the Mann–Whitney U test. To enter the regression model, the variables were selected using forward selection and were included if *p* < 0.2.

The determinants of UI that were significant in the logistic regression models were further used in receiver operating characteristic (ROC) analyses. For each ROC analysis, the area under the curve (AUC) and associated 95% confidence interval (CI) were calculated. Additionally, the cut-off values with the highest level of sensitivity and specificity were established. The statistical analysis was performed using Statistica PL software (version 12.0; StatSoft, Inc., Tulsa, OK, USA) and MedCalc Statistical Software (version 16.8; MedCalc Software BVBA, Ostend, Belgium).

### 2.4. Ethical Considerations

This study was conducted in accordance with the Helsinki Declaration, and all patients gave their consent to participate in the study. The local bioethics committee approved the study protocol.

## 3. Results

### 3.1. Study Population

Sonographic images of adequate quality could not be obtained in one patient, who was therefore excluded from the study. Thus, 71 women, including 34 (47.2%) with UI and 37 (52.8%) without UI (control group), were enrolled in the study. The two groups did not differ in terms of age, body height, body weight, age at menopause, pthe resence of POP (POPQ 1+), or the mode of childbirth/delivery, as presented in Table 1.

### 3.2. Static and Dynamic (Functional) Angles and Diameters

We performed ultrasonographic assessments of all patients at rest, during the Valsalva maneuver, and during squeezing. Patients with UI showed a lower BSD during the Valsalva maneuver (15 vs. 23 mm, *p* = 0.03), a larger BSD difference between rest and the Valsalva maneuver (16 vs. 7 mm, *p* = 0.002), a larger alpha angle during the Valsalva maneuver (125.5° vs. 95°, *p* = 0.045), a larger alpha angle difference between the Valsalva maneuver and rest (53.5° vs. 31°, *p* = 0.03), a larger beta angle during the Valsalva maneuver (38° vs. 22°, *p* = 0.026), a larger gamma angle during the Valsalva maneuver (134° vs. 118°, *p* = 0.019), and a larger gamma angle difference between the Valsalva maneuver and rest (41° vs. 21°, *p* = 0.005). The detailed data for static and dynamic measurements are shown in Table 2 and are visually depicted in Figure 1 and Figure 2.

### 3.3. Urethral Diameters and Specific Sonographic Symptoms

Additionally, our approach for ultrasonography assessment included urethral measurements at three levels. Qualitatively, we assessed the presence of the “funneling” and “total urethral rotation” signs at rest and during the Valsalva maneuver. We additionally assessed the presence of urethral rotation. The urethra was more dilated at all three levels in all patients with UI, as shown in Table 3 and graphically presented in Figure 3 and Figure 4. Additionally, the funneling sign during the Valsalva maneuver was more frequent in patients with UI (85.3% vs. 21.6%, *p* < 0.001).

### 3.4. Utility of Ultrasonographic Measurements

Next, we assessed all identified relevant parameters for their utility in predicting UI. Using a logistic regression model, we found that two of the three planned models showed statistical significance (Table 4). These results were unaltered after adjustment for age, body mass index, or the presence of POPQ, as shown in Appendix A. Among measurements based on US anatomical angles and diameters, the difference in BSD obtained at rest and during the Valsalva maneuver and the mean urethral diameter were independent predictors of UI. Each 1 mm increase in the BSD increased the odds of UI by 15%, while each 1 mm increase in the urethral diameter increased these odds by 328%. We also found that, among specific phenomena, the occurrence of the funneling sign during the Valsalva maneuver was strongly associated with UI, as shown in Table 4. Finally, we assessed these two variables using ROC curves to establish cut-off points for predicting UI. For the BSD difference between rest and the Valsalva maneuver, the optimal cut-off point was >8 mm, which showed high sensitivity (91.2%) and moderate specificity (56.8%) with a good area-under-curve value of 0.72 (*p* = 0.001). The ROC curve of the central urethral diameter indicated that the optimal cut-off point was >6 mm, which showed high sensitivity (82.4%) while maintaining moderate specificity (73%) with a good area-under-curve value of 0.84 (*p* < 0.001). When we compared the AUCs of these two ROC curves, we found that both measurements of mean diameter and BSD difference between rest and the Valsalva maneuver were comparable (AUC difference, 0.12; *p* = 0.17). ROC analyses are presented in Figure 5.

## 4. Discussion

The loss of anatomical support for the bladder neck and proximal urethra, as well as for the distal urethra, is an important factor associated with UI. The hypermobility of the ureterovesical junction is an important clinical determinant of SUI [2,3]. Over the past few decades, the Q-tip test and cystourethrography have been used to detect abnormal mobility in this part of the urinary tract. However, the efficiency of these methods is not satisfactory. Many years of research and observations have revealed the importance of identifying and repairing all existing defects in the support apparatus of the bladder, urethra, uterus, and vagina to achieve high treatment efficacy. Otherwise, the therapy may be ineffective or may lead to additional symptoms caused by further damage at other pelvic sites as a result of limiting the correction to the most evident disorders.

Our study compared transperineal US findings for the pelvic floor in women showing UI with or without concomitant POP with the corresponding findings in women with or without POP who did not show UI. Standard parameters and new measurements were utilized for the assessment of ultrasound images, and their use was supported by the experience gained from diagnosing women with UI. The two groups of patients did not differ significantly in terms of age, body mass index, or number of pregnancies.

In this study, the mean BSD during the Valsalva maneuver was significantly higher in the women with UI. Additionally, this parameter was shown to exhibit the highest sensitivity among all assessed distance and angular measurements. Similar conclusions were obtained by Turkoglu et al. [4]. Dietz reported that a BSD difference > 25 mm between the measurements obtained at rest and during the Valsalva maneuver indicated the hypermobility of the bladder neck [5]. The importance of this parameter in diagnostic urogynecological ultrasound assessments was also confirmed by Pregazzi et al., who reported a BSD cut-off value of 26 mm [6]. The practical value of BSD was described by Torella, who showed that reducing the difference in this distance measurement at rest and during exertion was associated with therapeutic success [7].

Other measurements objectifying ultrasound assessments of the anterior compartment and playing an important role in the diagnosis of SUI included the angles between the appropriate sections of the urethra, the pubic symphysis, and the bladder wall. These angles are mathematical representations of spatial positional changes, especially those of the urethra. The utility of the so-called symptom of total urethral rotation, defined by a significant disappearance of the characteristic hypoechoic image of the interior of the urethra during the Valsalva maneuver and the funnel sign, was also assessed in the present study.

The alpha angle, described by Sarnelli, represents the degree of the inclination of the proximal urethral axis in relation to the *x*-axis of the symphysis pubis (central line) [8]. In healthy women, it ranges from 80° during pelvic floor squeezing to 100° during the Valsalva maneuver; however, in women with SUI, this angle ranges from 90° to 120° [6]. Our study confirmed these differences. Similar results were obtained by Wasan Ismail Al-Saadi, who also used the transperineal technique [9]. Analogous relationships were demonstrated by measuring this angle in images obtained with a vaginal transducer [10,11,12]. Suburethral slings, which are currently used as a treatment of choice for SUI, create an artificial support to replace the damaged part of the intrapelvic fascia and thus eliminate the observed pathological changes in BSD and the alpha angle during exertion [6].

The beta angle, which is formed by lines parallel to the proximal and distal urethral axes, was another analyzed parameter. The posterior urethrovesical angle has also been described as the beta angle in the literature. The mean beta angle during the Valsalva maneuver was significantly greater in the group with SUI than in the control group (38° vs. 22°) in this study. In the literature, the behavior of this angle was previously analyzed by Pregazzi et al., who compared 23 women with urodynamically confirmed SUI and 50 patients without UI and used ROC curves to establish a cut-off point of 14° that indicated incontinence. Pregazzi et al. also reported that the beta angle showed more sensitivity than the BSD for the detection of SUI in women. The beta angle obtained in this study, which is representative of patients with incontinence, was different from that established by Pregazzi et al. [6].

The results described by DeLancey imply that the distal urethra is less mobile than the proximal part, and its main role in UI is to maintain adequate intraurethral pressure, which is generated by the surrounding muscles, i.e., the urethral compressor muscle and the urethrovaginal sphincter. This urethral segment lies within the urogenital diaphragm [13], providing a theoretical basis for concluding that the beta angle between the lines tangential to both parts of the urethra should be greater in cases of SUI since its proximal part may remain more mobile during straining or coughing. This was confirmed in the present study.

The gamma angle, which is measured between the neck of the bladder and the lower border of the symphysis pubis, may be analogous to the alpha angle since the two angles share a reference line: for the gamma angle, the second arm is a line running toward the bladder neck, and, for the alpha angle, it is a line tangential to the axis of the proximal urethra. The lines show similar movements with changes in the position of the urethro–vesical junction. Although the present study showed no statistically significant differences in the gamma angle measured at rest and during perineal squeezing, differences were observed during the Valsalva maneuver. This was also reflected after subtracting the gamma angle value during exertion and at rest. These results clearly confirm the relationship between hypermobility of the bladder neck and SUI. No reflections on the role of the gamma angle in UI have been found in the available literature.

The RVA, another parameter assessed in this study, is measured between the proximal urethral axis and the line tangential to the lowest part of the posterior wall of the bladder. An RVA between 90° and 120° is considered normal [14]. However, the measurement of this angle provides no new value to the ultrasound diagnosis of UI due to the lack of significant differences between the groups [15]. RVA was only slightly higher in the SUI group during the measurement at rest and during perineal squeezing. However, Sendag et al. reported a similar but statistically significant relationship [16], while other authors reported that RVA > 120°, both at rest and during squeezing, was correlated with UI [17,18]. On the other hand, Cendrowski et al. described that the ultrasound measurement of the RVA is a non-invasive method of objectifying the diagnosis of SUI in women and that this measurement may be a simple method for predicting the effectiveness of surgical treatment [19]. The RVA is undoubtedly an important parameter for evaluating the urethrovesical junction, particularly the status of the surrounding tissues, on which its values depend.

The sonographic funneling symptom is defined as the opening of the proximal urethra on exertion [20]. Additionally, US can visualize the outflow of urine through the internal urethral opening on straining, and color Doppler can provide a more distinct visualization. Our findings, similar to the results obtained by other authors, clearly indicate the presence of the funneling sign in women with UI [21,22,23]. The funneling sign most likely indicates internal urethral sphincter insufficiency. This pathology may occur in isolation or overlap with other causes of UI, such as the hypermobility of the bladder neck. In our opinion, when the funneling effect predominates over other abnormalities, a tension-free vaginal tape is more effective than transobturator tape.

The beneficial effect of retropubic sling placement on treatment outcomes in women with sonographic funneling has been demonstrated by Harms. In a group of 171 women who qualified for tension-free vaginal tape insertion due to SUI, the funneling rate dropped from 37.2% pre-surgery to 17.3% post-surgery. Correct postoperative urinary continence was observed in 57.5% of women with sphincter insufficiency, in comparison with 96.2% of women with normal sphincter function [24].

The new sonographic parameters in the diagnosis of UI that were developed on the basis of our own experience in urogynecological ultrasonography were the sum of the urethral widths measured at the level of the internal sphincter, the middle part and the external sphincter of the urethra, and the presence of total urethral rotation. The sum of the widths for all three urethral dimensions, referred to as the total urethral width, was significantly greater in the UI group than in the control group, suggesting that excessive periurethral relaxation at various levels and the consequent urethral widening may be one of the mechanisms leading to UI. Another evaluated parameter, total urethral rotation, showed a significant loss of urethral lumen during the Valsalva maneuver. This may result from urethral movement in all three dimensions, including its twisting around its axis. Urethral rotation was detected in 61.8% of SUI patients in comparison with 46.0% of controls. However, the differences were not statistically significant (*p* = 0.18).

This study also highlighted the significant utility of several sonographic parameters in the diagnosis of SUI. Notably, the differences between our findings and those presented by other authors may have been influenced by factors such as bladder filling and the patient’s position during the examination. Our observations indicated that the amount of urine affects RVA measurement. The potential lowering of the bladder wall from the side of the vagina increases with increased bladder filling. The bladder filling volume in this study was maintained between 100 mL and 250 mL to avoid its influence on the results. The effectiveness of the Valsalva maneuver is another factor that may affect transperineal measurements. At this point, the co-activation of the levator ani muscle and the displacement of the ultrasound transducer that occur while squeezing should be taken into account [25]. On the other hand, the intravaginal application of the transducer stabilizes the pelvic structures and affects their actual mobility. Thus, the methodology for transperineal ultrasonography requires standardization not only in terms of the parameters measured, but also for proper patient preparation.

The clear advantages of the ultrasonographic analysis of the urethra are that no X-rays are involved, no contrast medium is necessary, there is limited invasiveness, the technique is low-cost, it is an office procedure, and it has good anatomical accuracy. The disadvantages are the learning curve for the practitioner, that it is operator-dependent, and that there is presently limited scientific evidence (no randomized clinical trial/level 1 evidence) to support the use of the method. In the future, studies should attempt to obtain such high-level evidence regarding the use of urethral ultrasonography. In principle, applications of transperineal sonography could be expanded to other conditions, such as, for example, interstitial cystitis/bladder pain syndrome and conditions where, at the moment, invasive diagnostics are still largely used and only a few non-invasive tests are reliable. The use of transperineal ultrasonography in such conditions would limit patient pain and discomfort [26].

Our study had multiple strengths. Unlike the transvaginal approach, transperineal ultrasound does not exert pressure on the examined structures, precluding a modulating effect on the obtained results. Moreover, transperineal ultrasound is an inexpensive and accessible method that can confirm the clinical manifestations of SUI and serves as an alternative to modalities such as urodynamic studies, thereby supporting the decision-making process in terms of qualification for surgical treatment. Another advantage of this study was that ultrasound examination was performed by one doctor trained in this technique. This ensured that all the patients were examined by the same person, reducing the risk of measurement bias. In the analyses, we achieved very high sensitivity in detecting SUI from mean urethral diameter.

However, the study also had several limitations: This was a single-center study and the number of patients examined was small.

## 5. Conclusions

Transperineal ultrasonography is a useful non-invasive tool for the reliable diagnosis of patients with UI. The difference in the BSD at rest and during the Valsalva maneuver, the funneling sign during the Valsalva maneuver, and the mean urethral diameter are the three anatomical and functional parameters of particular clinical relevance. Between them, the mean urethral diameter was superior to the others and appears to be particularly useful in clinical practice. Evaluations based on mean urethral diameter > 9 mm showed good sensitivity (82.1%) along with an excellent AUC in ROC analysis. The transperineal ultrasonographic evaluation of SUI will benefit clinicians in the future as a technique that is fast, cost-effective, and easy to teach [27]. In conclusion, we propose transperineal ultrasonography as a reliable and effective non-invasive method for the accurate diagnosis of women with SUI. More evidence on this topic and more data from studies with larger patient groups need to be collected in the future.

## Figures and Tables

**Figure 1 diagnostics-14-02549-f001:**
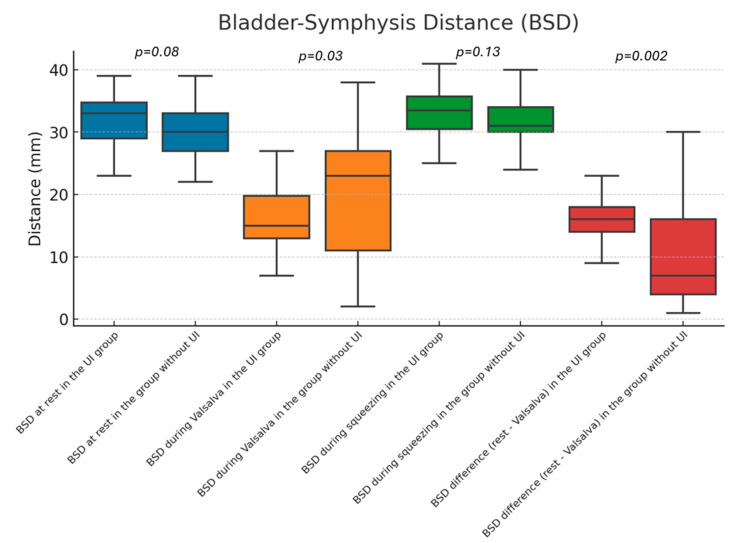
Bladder–symphysis distance in patients with and without urinary incontinence.

**Figure 2 diagnostics-14-02549-f002:**
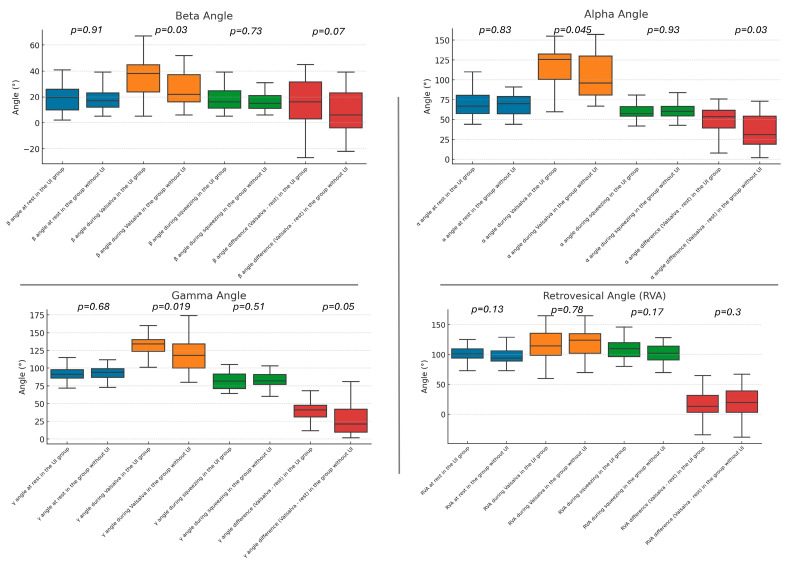
Angles assessed in ultrasonography in patients with and without urinary incontinence.

**Figure 3 diagnostics-14-02549-f003:**
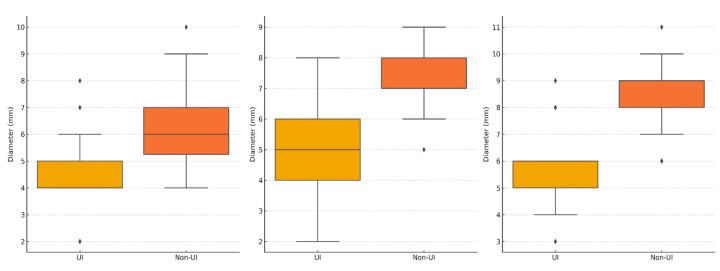
Urethral diameters. Data are presented as median and interquartile range. The box plots represent the distribution of urethral diameters measured in millimeters for individuals with and without urinary incontinence (UI). Comparisons between groups were made using the Mann–Whitney U test.

**Figure 4 diagnostics-14-02549-f004:**
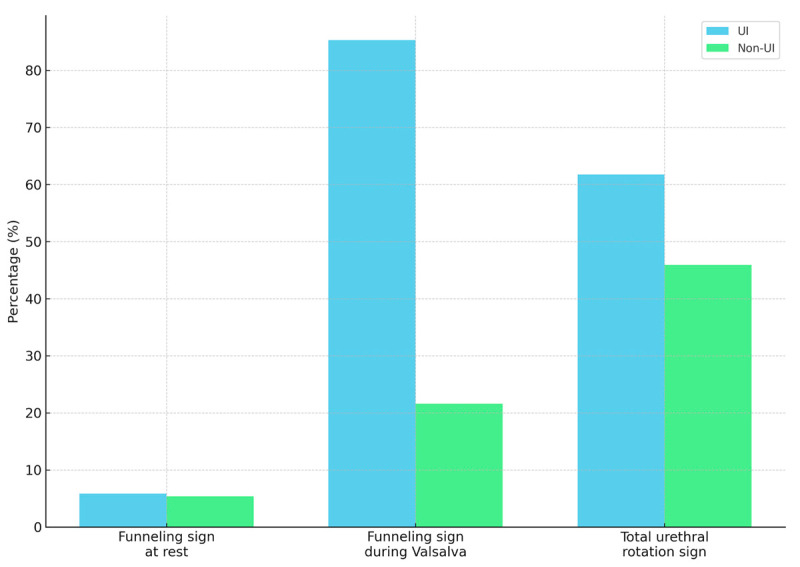
Funneling and urethral rotation signs. Bars represent the percentage of individuals exhibiting the funneling sign at rest, during the Valsalva maneuver, and total urethral rotation. Comparisons between groups (urinary incontinence [UI] vs. non-UI) were made using the chi-square test.

**Figure 5 diagnostics-14-02549-f005:**
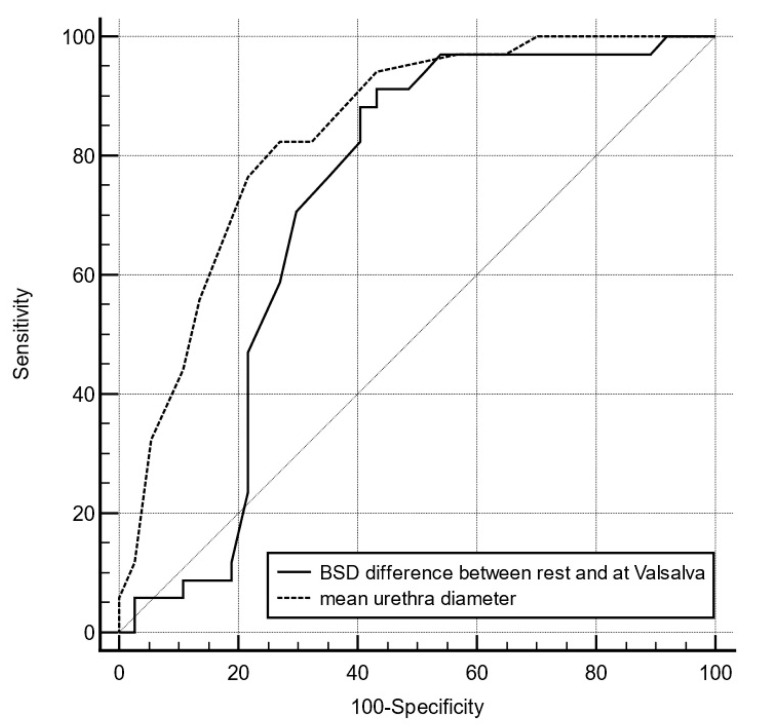
Receiver operating characteristic curve for the identification of urinary incontinence. We assessed variables shown to be significant in logistic regression: BSD difference between rest and during the Valsalva maneuver with a cut-off point of >8 mm (area under the curve, 0.71; sensitivity, 91.2%; specificity, 56.8%; *p* = 0.001) and mean urethral diameter with a cut-off point of >6 mm (area under the curve, 0.84; sensitivity, 82.1%; specificity, 73%; *p* < 0.001). BSD, bladder–symphysis distance.

**Table 1 diagnostics-14-02549-t001:** Clinical characteristics of the study participants.

	UI (*n* = 34)	Non-UI Control Group (*n* = 37)	*p*
Age (years)	55 (47–63)	59 (49–66)	0.23
BMI (kg/m^2^)	25.6 (21.9–29.2)	25.9 (24.9–29.1)	0.45
Post-menopausal	19 (55.9%)	25 (70.2%)	0.311
Number of deliveries (median, IQR)	2 (2–3)	2 (2–3)	0.38
C-sections (median)	0	0	0.55
Pelvic organ prolapse (*n*, %)	27 (79.4%)	25 (67.6%)	0.26
Arterial hypertension (*n*, %)	15 (44.1%)	13 (35.1%)	0.44
Diabetes (*n*, %)	4 (11.8%)	2 (5.4%)	0.34
Chronic obstructive pulmonary disease (*n*, %)	2 (5.9%)	1 (2.7%)	0.51
Urinary tract infection in the last year	8 (23.5%)	11 (29.7%)	0.56
Total hysterectomy	2 (5.9%)	1 (2/7%)	0.51

BMI, body mass index; IQR, interquartile range; UI, urinary incontinence. The data are presented as median and IQR and were compared using the Mann–Whitney U test.

**Table 2 diagnostics-14-02549-t002:** Static and dynamic (functional) parameters measured in transperineal ultrasonography for the presence of urinary incontinence.

	UI Group (*n* = 34)	Non-UI Group (*n* = 37)	*p*
BSD at rest (mm)	33 (29–35)	30 (27–33)	0.08
BSD during the Valsalva maneuver (mm)	15 (13–20)	23 (11–27)	0.03
BSD during squeezing (mm)	33.5 (30–36)	31 (30–34)	0.13
BSD difference between rest and the Valsalva maneuver (mm)	16 (14–18)	7 (4–16)	0.002
α angle at rest (°)	67 (57–81)	70 (57–79)	0.83
α angle during the Valsalva maneuver (°)	125.5 (100–133)	96 (81–130)	0.045
α angle during squeezing (°)	57.5 (54–67)	60.5 (54–67)	0.93
α angle difference between the Valsalva maneuver and rest (°)	53.5 (39–62)	31 (19–54)	0.03
β angle at rest (°)	19.5 (10–26)	17 (12–23)	0.91
β angle during the Valsalva maneuver (°)	38 (23–45)	22 (16–37)	0.026
β angle during squeezing (°)	16 (11–25)	15 (11–21)	0.73
β angle difference between the Valsalva maneuver and rest (°)	16 (3–32)	6 (−4–23)	0.07
γ angle at rest (°)	91.5 (86–98)	94 (87–99)	0.68
γ angle during the Valsalva maneuver (°)	134 (122–141)	118 (100–134)	0.019
γ angle during squeezing (°)	81.5 (70–92)	82 (77–91)	0.51
γ angle difference between the Valsalva maneuver and rest (°)	41 (31–48)	21 (10–42)	0.005
RVA at rest (°)	101 (94–110)	94 (89–106)	0.13
RVA during the Valsalva maneuver (°)	114.5 (98–136)	124 (102–135)	0.78
RVA during squeezing (°)	110 (96–120)	102 (91–114)	0.17
RVA difference between the Valsalva maneuver and rest (°)	13.5 (3–32)	20 (3–39)	0.3

BSD, bladder–symphysis distance; RVA, retrovesical angle; UI, urinary incontinence. The data are presented as median and interquartile range and were compared using the Mann–Whitney U test.

**Table 3 diagnostics-14-02549-t003:** Measurements of static urethral diameters at three levels and static and dynamic assessments of the following signs: funneling and total urethral rotation at rest and during the Valsalva maneuver.

	UI (*n* = 34)	Non-UI Control Group (*n* = 37)	*p*
Internal urethral sphincter diameter (mm)	6 (5–7)	4 (4–5)	<0.001
Diameter of the central part of the urethra (mm)	7 (7–8)	5 (4–6)	<0.001
External urethral sphincter diameter (mm)	9 (8–9)	6 (5–6)	<0.001
Funneling sign (*n*, %)	2 (5.9%)	2 (5.4%)	0.93
Funneling sign during the Valsalva maneuver (*n*, %)	29 (85.3%)	8 (21.6%)	<0.001
Total urethral rotation sign (*n*, %)	21 (61.8%)	17 (46%)	0.18

UI, urinary incontinence. The data are presented as median and interquartile range and were compared using the Mann–Whitney U or χ^2^ test.

**Table 4 diagnostics-14-02549-t004:** Significant results of logistic regression analyses for the detection of urinary incontinence with two different approaches.

	OR	95% CI	*p*-Value for the Variable
Model 1: Ultrasound parameters based on anatomical angles and diameters (BSD; alpha, beta, gamma, and retrovesical angles; urethral diameter).			
BSD difference between rest and the Valsalva maneuver (mm)	1.15	1.05–1.27	0.0039
Mean urethral diameter (mm)	4.28	2.07–8.83	0.0001
Model 2: Presence of specific ultrasound qualitative signs			
Funneling sign during the Valsalva maneuver	21	6.1–71.9	<0.0001

Model 1 used anatomical diameters and angles measured in ultrasound assessments (*p* = 0.001 for the model, R^2^ = 0.57). Model 2 used specific qualitative ultrasound symptoms at rest or during the Valsalva maneuver (*p* < 0.0001 for the model, R^2^ = 0.48). Forward stepwise selection was used; variables were included if the initial *p*-value was 0.1 and removed if *p* > 0.2. BSD, bladder–symphysis distance; CI, confidence interval; OR, odds ratio.

## Data Availability

The data presented in this study are available on request from the corresponding author.

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
