# Peer review of "Enhanced Non-Invasive Diagnosis of Female Urinary Incontinence Using Static and Functional Transperineal Ultrasonography"

_diagnostics, 2024, doi:10.3390/diagnostics14222549_

Round 1

Reviewer 1 Report

Comments and Suggestions for Authors

Dear authors, I would like to thank you for the care and meticulousness you have shown in your article. Observing and evaluating the studies I have been doing on urinary incontinence for years and the developing technological developments makes me extremely happy. 

Below, I will give you a few suggestions and recommendations that will strengthen your article;

1- Strengthening the discussion with a sentence similar to the sample sentence below and a citation will make the article more compact. Adding a sentence like 'Compared to urodynamic evaluation or non-invasive physical examination, which are alternative invasive diagnostic methods, transperineal ultrasonographic evaluation will benefit clinicians in the future as a technique that is both fast, cost-effective and easy to teach' will make it stronger. 'Okuyan, E., Cakir, C., Gunakan, E., Ozkaya, E., Kucukozkan, T., & Sefik Ozyurek, E. (2021). Comparison of treatment results of urinary incontinence verified/or not verified with urodynamic evaluation by using UDI-6, IIQ-7 questionnaire forms . Annals of Medical Research, 27(8), 2094–2098. Retrieved from https://www.annalsmedres.org/index.php/aomr/article/view/905'.

2- Adding the sentence containing the roc analysis you wrote in the Conclusion subheading just before the sentence mentioning the limitations of the discussion will make the article more compact.

3- The language of the article flows well and the conclusion is effectively organized.

4-There are very few well-designed articles in the literature about non-invasive new generation diagnostic tests, and this article will contribute to the literature. The design is well constructed and the article is strengthened with the comparative group. Although the groups are not homogeneous due to article limitations, the groups can be homogenized in future studies.

Author Response

Reviewer #1

Dear authors, I would like to thank you for the care and meticulousness you have shown in your article. Observing and evaluating the studies I have been doing on urinary incontinence for years and the developing technological developments makes me extremely happy. 

Below, I will give you a few suggestions and recommendations that will strengthen your article;

Response: Thank you very much for all your valuable comments. We greatly appreciate that, as a person following research on urinary incontinence, you recognize the value of our work. We have provided responses to all your comments below.

1- Strengthening the discussion with a sentence similar to the sample sentence below and a citation will make the article more compact. Adding a sentence like 'Compared to urodynamic evaluation or non-invasive physical examination, which are alternative invasive diagnostic methods, transperineal ultrasonographic evaluation will benefit clinicians in the future as a technique that is both fast, cost-effective and easy to teach' will make it stronger. 'Okuyan, E., Cakir, C., Gunakan, E., Ozkaya, E., Kucukozkan, T., & Sefik Ozyurek, E. (2021). Comparison of treatment results of urinary incontinence verified/or not verified with urodynamic evaluation by using UDI-6, IIQ-7 questionnaire forms . Annals of Medical Research, 27(8), 2094–2098. Retrieved from https://www.annalsmedres.org/index.php/aomr/article/view/905'.

Response: Thank you for this suggestion. We have added the proposed sentence, which will certainly strengthen the discussion. We confirmed the presence of stress urinary incontinence based on our own questionnaire, which we attach as Appendix 1. We have added a sentence on this subject in the Methods section of the manuscript.

2- Adding the sentence containing the roc analysis you wrote in the Conclusion subheading just before the sentence mentioning the limitations of the discussion will make the article more compact.

Response: Thank you for this suggestion. To make the article more compact we have added this sentence as you suggest.

3- The language of the article flows well and the conclusion is effectively organized.

Response: Thank you very much. The text of the article was edited and certified by Oxford Science Editing Ltd.

4-There are very few well-designed articles in the literature about non-invasive new generation diagnostic tests, and this article will contribute to the literature. The design is well constructed and the article is strengthened with the comparative group. Although the groups are not homogeneous due to article limitations, the groups can be homogenized in future studies.

Response: Thank you very much for appreciating our contribution to the attempt to evaluate this diagnostic method for SUI. We hope that work such as ours will allow for better standardization and acceptance in the diagnosis of pelvic floor disorders and urinary incontinence.

Reviewer 2 Report

Comments and Suggestions for Authors

This study analyses the results of a group of 34 women (+ 37 controls), on the role of ultrasound imaging in the diagnosis of stress urinary incontinence, with the goal of providing indications for clinical practice. This is a challenging goal, given the specific difficult topic, the scarce available evidence, the paucity of the data coming from small patients’ group, and the limited expertise with the analysed diagnostic exam, transperineal ultrasonography of the urethra. This paper is therefore, in principle, interesting and well needed by the urological community, potentially able to shed some light and offer some valuable information in the field of non-invasive diagnosis of urinary incontinence. This manuscript needs a minor number of changes and ameliorations before being accepted for publication in Diagnostics. The changes needed are detailed in the text below for the Authors.

TITLE: to make it more accurate, I would change the title:

ENHANCED NON-INVASIVE DIAGNOSIS OF FEMALE URINARY INCONTINENCE USING STATIC AND FUNCTIONAL TRANSPERINEAL ULTRASONOGRAPHY

DISCUSSION:

Page 11, third line of the Discussion: “Hypermobility of the ureterovesical junction”: did you mean urethro-vesical junction?

Page 12, third paragraph of the Discussion: “In this study, the mean BSD during the Valsalva maneuver was significantly higher in the women with UI. The mean BSD during the Valsalva maneuver also differed from that obtained at rest”: please re-phrase, unclear.

Page 14, first paragraph: “The funneling sign most likely indicates internal urethral sphincter insufficiency etc. etc.” These statements should be reported as the Authors’ opinion, and not as facts, and the reported not recent citation (from 2006, cit. n. 24), does not provide clear-cut evidence.  So, the end of the paragraph should be changed, for example “(it is the Authors’ opinion that) when the funneling effect predominates over other abnormalities, a tension-free vaginal tape could me more effective than a trans-obturator tape, although there is no clear-cut evidence about this”.

Clear advantages of the ultrasonography analysis of the urethra are: no X-rays involved, no contrast-medium necessary, limited invasiveness, low cost, office procedure, good anatomical accuracy. Disadvantages: learning curve, operator-dependent; limited scientific evidence and limited number of studies available, with small patients’ groups. No randomised clinical trials, no level 1 evidence at the moment available. These concepts should be clearly stated in the Discussion and then re-phrased synthetically in the Conclusions.

Moreover, the Authors can suggest that the applications of transperineal sonography could be expanded to other conditions, such as for example interstitial cystitis/bladder pain syndrome, conditions where at the moment invasive diagnostics are still largely utilised, and only a few non-invasive tests are reliable for applied at current; transperineal ultrasonography could be an important non-invasive test in this condition, limiting discomfort and pain to the patients (add reference: Morlacco A., et al: UROLOGY 144: 106−110, 2020).

CONCLUSIONS

Transperineal ultrasonography is a useful non-invasive tool for reliable diagnosis of patients with urinary incontinence. The difference in the BSD at rest and during the Valsalva maneuver, the funneling sign during the Valsalva maneuver and the mean urethral diameter are the three anatomical and functional parameters of particular clinical relevance. Between them, the mean urethral diameter was superior to the others and appears to be particularly useful in clinical practice. Evaluations based of mean urethral diameter >9mm showed good sensitivity (82.1%) along with an excellent AUC in the ROC analysis. In conclusion, we propose transperineal ultrasonography as a reliable and effective non-invasive methodology for accurate diagnosis of women with stress urinary incontinence. More evidence on this topic and more data from studies with larger patients’ groups need to be collected in the next future.

Author Response

Reviewer #2:

This study analyses the results of a group of 34 women (+ 37 controls), on the role of ultrasound imaging in the diagnosis of stress urinary incontinence, with the goal of providing indications for clinical practice. This is a challenging goal, given the specific difficult topic, the scarce available evidence, the paucity of the data coming from small patients’ group, and the limited expertise with the analysed diagnostic exam, transperineal ultrasonography of the urethra. This paper is therefore, in principle, interesting and well needed by the urological community, potentially able to shed some light and offer some valuable information in the field of non-invasive diagnosis of urinary incontinence. This manuscript needs a minor number of changes and ameliorations before being accepted for publication in Diagnostics. The changes needed are detailed in the text below for the Authors.

TITLE: to make it more accurate, I would change the title:

 ENHANCED NON-INVASIVE DIAGNOSIS OF FEMALE URINARY INCONTINENCE USING STATIC AND FUNCTIONAL TRANSPERINEAL ULTRASONOGRAPHY

Response: Thank you for your suggestion, we have changed the title to your recommendation.

DISCUSSION:

Page 11, third line of the Discussion: “Hypermobility of the ureterovesical junction”: did you mean urethro-vesical junction?

 Response: Thank you for spotting this error – it has been corrected.

Page 12, third paragraph of the Discussion: “In this study, the mean BSD during the Valsalva maneuver was significantly higher in the women with UI. The mean BSD during the Valsalva maneuver also differed from that obtained at rest”: please re-phrase, unclear.

Response: Thank you for pointing this out. To clarify the message, we have removed the second sentence.

Page 14, first paragraph: “The funneling sign most likely indicates internal urethral sphincter insufficiency etc. etc.” These statements should be reported as the Authors’ opinion, and not as facts, and the reported not recent citation (from 2006, cit. n. 24), does not provide clear-cut evidence.  So, the end of the paragraph should be changed, for example “(it is the Authors’ opinion that) when the funneling effect predominates over other abnormalities, a tension-free vaginal tape could me more effective than a trans-obturator tape, although there is no clear-cut evidence about this”.

Response: Thank you for noticing this inaccuracy. We have changed this sentence so that the claim is now stated as the authors’ opinion, and we have removed the citation.

Clear advantages of the ultrasonography analysis of the urethra are: no X-rays involved, no contrast-medium necessary, limited invasiveness, low cost, office procedure, good anatomical accuracy. Disadvantages: learning curve, operator-dependent; limited scientific evidence and limited number of studies available, with small patients’ groups. No randomised clinical trials, no level 1 evidence at the moment available. These concepts should be clearly stated in the Discussion and then re-phrased synthetically in the Conclusions.

 Response: Thank you for this advice, we have incorporated these points in the Discussion.

Moreover, the Authors can suggest that the applications of transperineal sonography could be expanded to other conditions, such as for example interstitial cystitis/bladder pain syndrome, conditions where at the moment invasive diagnostics are still largely utilised, and only a few non-invasive tests are reliable for applied at current. transperineal ultrasonography could be an important non-invasive test in this condition, limiting discomfort and pain to the patients (add reference: Morlacco A., et al: UROLOGY 144: 106−110, 2020).

Response: We fully agree with this comment. Originally, we did not want to include this comment so as not to prolong the discussion, but as recommended, we have added it along with the citation.

CONCLUSIONS

Transperineal ultrasonography is a useful non-invasive tool for reliable diagnosis of patients with urinary incontinence. The difference in the BSD at rest and during the Valsalva maneuver, the funneling sign during the Valsalva maneuver, and the mean urethral diameter are the three anatomical and functional parameters of particular clinical relevance. Between them, the mean urethral diameter was superior to the others and appears to be particularly useful in clinical practice. Evaluations based on mean urethral diameter >9 mm showed good sensitivity (82.1%) along with an excellent AUC in ROC analysis. In conclusion, we propose transperineal ultrasonography as a reliable and effective non-invasive method for accurate diagnosis of women with SUI. More evidence on this topic and more data from studies with larger patient groups need to be collected in the future.

Response: Thank you. We have adjusted the Conclusions in line with your suggestions (while also incorporating a suggestion from Reviewer 1).